# Cross-sectional study on intention to be vaccinated against Coronavirus Disease 2019 (COVID-19) in Benin and Senegal: A structural equation modeling (SEM)

**Ibrahima Gaye**[1]*, **Valery Ridde**[2], **Elías Martinien Avahoundjea**[3], **Mouhamadou Faly Ba**[1], **Jean-Paul Dossou**[3], **Amadou Ibra Diallo**[1], **Adama Faye**[1]

**1** Institute of Health and Development (ISED), Cheikh Anta Diop University, Dakar, Senegal, **2** CEPED, IRD-University of Paris, ERL INSERM SAGESUD, Paris, France, **3** Center for Research in Human Reproduction and Demography (CERRHUD), Cotonou, Benin

* ibr.gaye@gmail.com

**Data Availability Statement:** Download data and analysis codes freely here: https://doi.org/10.6084/m9.figshare.24574432.v2

## Abstract

Vaccination is considered one of the solutions to the Coronavirus Disease 2019 (COVID-19) pandemic. However, a small proportion of the population were fully vaccinated in Benin (20.9%) and Senegal (7.6%) by December 2022. This study explores the determinants of intent to vaccinate. This was a cross-sectional, descriptive, and analytical study of 865 Beninese and 813 Senegalese aged 18 years and older. Marginal quota sampling by age, gender and region was adopted. Data collection, using a survey instrument based on the Random Digit Dialing method, was conducted from December 24, 2020, to January 16, 2021, in Senegal and from March 29 to May 14, 2021, in Benin. The questionnaire used the Theory of Planned Behavior and the Health Belief Model. The influence of factors was assessed using a structural equation model based on a diagonally weighted least squares estimator to account for ordered categorical data (Likert scales). In Benin and Senegal, the intention to vaccinate against COVID-19 is influenced by distinct factors. In Benin, social influence ($\beta = 0.42$, $p = 0.003$) and perception of vaccine safety ($\beta = -0.53$, $p<0.001$) play pivotal roles, suggesting those socially influenced have a higher vaccination intention. In Senegal, vaccination intentions are primarily driven by positive attitudes towards the vaccine ($\beta = 0.65$, $p = 0.013$) and social influence ($\beta = 0.25$, $p = 0.048$). This underscores the importance of individual beliefs, personal perceptions, and supportive social contexts in decision-making. Notably, positive vaccination attitudes and perceptions in both countries are strongly tied to increased social influence. While nuances exist between Benin and Senegal regarding factors influencing COVID-19 vaccination intentions, both nations underscore the pivotal roles of social influence and individual vaccination perceptions. Emphasizing trust in vaccine safety and promoting positive attitudes through effective communication are crucial for enhancing vaccination uptake in these West African countries.

**Funding:** This research is part of the Support to the African Response to the COVID-19 Epidemic (ARIACOV) program funded by the French Development Agency (AFD). The funders had no role in study design, data collection and analysis, decision to publish, or preparation of the manuscript.

**Competing interests:** The authors have declared that no competing interests exist.

## Introduction

On March 11, 2020, the WHO declared the COVID-19 outbreak a global pandemic and urged states to take immediate measures to limit the infection's spread and ensure compliance with international health regulations (2005) [1]. Consequently, many countries decided to prohibit public gatherings, implement social distancing, and especially enforce containment measures. Meanwhile, vaccines against COVID-19 were being developed [2]. As a result, the WHO authorized [3], for emergency use, the Pfizer (December 31, 2020), AstraZeneca (February 15, 2021), Johnson & Johnson (March 12, 2021), and Sinopharm (May 7, 2021) vaccines.

The development and distribution of vaccines against COVID-19 are considered by the WHO and governments as an effective solution to limit the ability of the pathogen's ability to spread [4]. Indeed, through so-called "herd", "indirect", or "group" immunitý [4], vaccination enables individuals and communities to remain protected and reduce the likelihood of an outbreak. For example, the governments of Benin and Senegal launched their COVID-19 vaccination campaigns on February 23, 2021, and March 29, 2021, respectively. However, as of December 4, 2022, only three countries in Africa have achieved the target of fully vaccinating 70% of their population according to the WHO Strategy [5]: Seychelles (76.7%), Liberia (79.9%) and Mauritius (86.0%). By December 2022, Benin has fully vaccinated 20.9% of its population, while Senegal (7.6%) has not yet exceeded the 10% threshold [5]. This low vaccination coverage observed in both countries does not eliminate the possibility of epidemic transmission, given the presence of a reservoir of unvaccinated individuals susceptible to infection. What motivates people in Benin and Senegal to accept vaccination?

Recent studies have shown that concerns about perceived safety and efficacy [6–8]. or lack of reliable information about vaccines [9, 10] were the main barriers to adoption of SARS-CoV-2 vaccines. However, these works use classical techniques with limitations for modeling vaccine intention or hesitancy, a complex decision-making process with multiple sources of influence [11, 12]. In contrast to these classical methods, structural equation models can meet the following conditions [13]: the ability to simultaneously handle several sets of observed explanatories and explained variables (hence the stage of causal relationships), the ability to analyze the links between the different dimensions and to consider errors at the level of measurement (reduction of psychometric biases), and finally, the ability of confirmatory applications.

Structural equation modeling (SEM) was adopted to better understand and interpret decision-making on vaccination against COVID-19 in Benin and Senegal.

## Materials and methods

### Study area

Senegal is in West Africa with 14 administrative regions. The population of Senegal in 2019 is estimated at 16,209,125 and is a quite an even split (female = 50.2% and male = 49.8%). The ratio of telephone numbers per person is 1.1. The proportion of people using a cell phone at least five times a day increased from 36.4% in 2014 to 73.5% in 2017 [14]. With a population of 11,496,140, 50.9% of whom are women, Benin is a West African country. In 2021, cell phone penetration was estimated at 101.8%, which corresponds to a mobile subscriber base of 12,731,782 [15].

### Type of study

This was a cross-sectional, descriptive, and analytical nationwide study remotely conducted via telephone from a call center. Data were collected from December 24, 2020, to January 16, 2021, for Senegal and from March 29 to May 14, 2021, for Benin.

### Study population

The study population consisted of the populations of Senegal and Benin, aged 18 years and above, proportionally distributed according to age, sex, and region.

### Sample size and sampling

To determine the sample size, simulations were conducted to illustrate the effect of the population size on the degree of confidence in the results using the following formula [16, 17]:

$$e = Z_{\alpha} * \sqrt{1 - \left(\frac{n}{(N-n)}\right)} \sqrt{\frac{p(1-p)}{n}} \tag{1}$$

- N = Parent population size

 n = Sample size

- p = Expected proportion in the population

- α = Confidence level

- $Z_{\alpha}$ = Value read from the standard normal distribution table

The simulations indicate that for a sample of 1000 individuals, the precision of the results is similar (approximately 3%) once the parent population size exceeds 100,000 (Table 1). The choice of P = 50% is because sampling error is greatest when the proportion is 50%, giving the largest sample size needed for a given precision [18].

A marginal quota survey was carried out on a target sample of 1,000 individuals [19]. This method is relevant in emergencies such as the COVID-19 pandemic with sample sizes below 3000 [19, 20]. An appropriate choice of quotas can reduce the estimate's variance and the magnitude of its confidence interval. If done rigorously, the quota sampling method can be as accurate as random sampling [20] or better if the sample size is small [21, 22]. The following variables were used to define the quotas: age, gender, and region [23]. The selection of these quota variables accounts for significant regional disparities in health practices, access to health

**Table 1. Simulations on the effects of population size and sample size on the level of precision.**

| N | n | P | α | e |
|---|---|---|---|---|
| 10 000 | 1000 | 50% | 5% | 2.9% |
| 50 000 | 1000 | 50% | 5% | 3.1% |
| 100 000 | 1000 | 50% | 5% | 3.1% |
| 200 000 | 1000 | 50% | 5% | 3.1% |
| 500 000 | 1000 | 50% | 5% | 3.1% |
| 1 000 000 | 1000 | 50% | 5% | 3.1% |
| 10 000 000 | 1000 | 50% | 5% | 3.1% |
| 100 000 000 | 1000 | 50% | 5% | 3.1% |

services, and attitudes towards vaccination that can influence individual decisions [24]. Furthermore, age is a critical factor since perceptions and behaviors towards health risks often differ between younger and older individuals, thus affecting the intention to get vaccinated [25]. Lastly, gender is a key variable due to the pronounced differences in social norms and roles that shape perceptions of vaccination and health-related decision-making [26]. Utilising these specific variables ensures a balanced and nuanced representation of the studied population, which is essential for the accuracy and relevance of our analyses concerning vaccination intent.

It's important to stress that this quota approach was adopted in this study not to guarantee regional representativeness but to ensure a balanced sample distribution across the different regions [27]. The aim was to include diverse responses without necessarily aiming for statistical representativeness by region. Our main concern was to avoid over-representing certain groups and ensure that our sample adequately covered the diversity of experiences and perspectives across the country.

The survey questionnaire was administered to a final sample of 813 individuals in Senegal and 865 individuals in Benin. Differences between the target sample and the final sample arose due to non-response from the targeted individuals or practical challenges during data collection (network issues, unavailability, etc.). To correct potential biases introduced by these differences, post-stratification using margin calibration was employed to adjust the data from the final sample, making it more representative of the target population. This is a statistical technique that enables more precise and reliable conclusions to be drawn from the final sample, despite the initial differences [28].

## Theoretical and conceptual frameworks

The design of the study questionnaire is grounded in two well-established theoretical models, deliberately selected for their direct relevance to the central research question at hand. These two foundational models are the Theory of Planned Behavior (TPB) and the Health Belief Model (HBM). [29–31].

The incorporation of the TPB allows us to explore the intricate interplay between individuals' attitudes, social influence, perceived behavioral control, and their intentions to engage in specific health-related behaviors. This model offers a comprehensive framework to examine how personal beliefs, social influences, and perceived control impact behavioral choices.

The utilization of the HBM enriches our understanding by addressing the perceptual and motivational aspects of health-related decisions. This model emphasizes individuals' perceptions of the severity of a health issue, their susceptibility to it, the perceived benefits of taking preventive actions, and the barriers that may hinder such actions. By integrating the HBM, we gain insights into the cognitive processes underlying health-related choices.

By strategically merging these two theoretical models into the questionnaire design, we aim to comprehensively capture the multidimensional aspects of individuals' decision-making processes regarding health behaviors. This combined approach allows us to explore not only the motivational factors behind these behaviors but also the cognitive evaluations that influence them.

Ultimately, the thoughtful incorporation of the TPB and the HBM into our questionnaire design enhances the depth and precision of our investigation, offering a robust framework for analyzing the factors driving health-related decisions in the context of our study [29, 30, 31].

## Description of scales and subscales

Vaccine intention for COVID-19 is the fundamental component of the SEM model. It is a manifest variable measured precisely by the following affirmative question: "I intend to get vaccinated against COVID-19". Responses were defined according to a 5-point Likert scale ("Strongly agree = 5", "agree = 4", "Neutral = 3", "disagree = 2", "Strongly disagree = 1").

This intention is based on the individual's willingness or interest in receiving the vaccine, typically influenced by their beliefs, knowledge, attitudes, and perceptions regarding the safety, effectiveness, and benefits of vaccination, available vaccine information, concerns about side effects, trust in health authorities, and other psychosocial and cultural factors [29, 32]. The following scales were considered for implementing the SEM model. Each subscale was measured on a 5-point Likert scale ("Strongly agree = 5", "agree = 4", "Neutral = 3", "disagree = 2", "Strongly disagree = 1").

## Covid-19 vaccine information seeking

This label accurately captured the underlying construct being assessed, which is the proactive seeking of information about the COVID-19 vaccine through various means, including regular information-seeking behavior, active efforts to better understand the vaccine, and reading information received through social media channels. The scale is assessed across three (3) subscales:

- **Regular Information Seeking About COVID-19 Vaccine**: This variable measures individuals' intention to stay informed about the COVID-19 vaccine through regular information-seeking behavior soon. It gauges their readiness to actively follow updates and developments regarding the vaccine.

- **Information seeking to better understand the coronavirus vaccine**: This variable evaluates individuals' inclination to seek information about the vaccine to enhance their understanding of it. Responses reflect their desire to acquire knowledge about the vaccine.

- **Reading Information Received About COVID-19 Vaccine via social media**: This variable assesses individuals' intent to actively engage with information about the COVID-19 vaccine that they receive through social media. It gauges their level of commitment to accessing information from this specific source.

The Cronbach's alphas reveal satisfactory reliability of the measurement scale for Benin ($\alpha$ = 0.72) and acceptable for Senegal ($\alpha$ = 0.68) [32].

## Perception of COVID-19 vaccine safety

This scale aims to evaluate individuals' perceptions regarding the safety of the COVID-19 vaccine. It consists of three statements, each representing a different aspect of this perception:

- **Belief in vaccine safety (reverse score):** This variable measures the degree to which individuals trust that those responsible for creating the COVID-19 vaccine will ensure its safety. The responses are often reverse scored so that higher scores indicate greater confidence in vaccine safety.

- **Perception of health risk**: This variable assesses individual's perception of whether the COVID-19 vaccine could pose a threat to their health. Responses indicate how much individuals believe the vaccine could endanger their health. Lower scores suggest a more negative perception of safety.

- **Concerns about side effects**: This variable gauges individual's concerns regarding potential side effects of the COVID-19 vaccine. Responses indicate how worried individuals are about vaccine side effects. Lower scores signify greater concerns.

Cronbach's alphas indicate acceptable reliability of the measurement scale in Benin ($\alpha$ = 0.70) and Senegal ($\alpha$ = 0.65) [32].

## Perceptions of COVID-19 vaccination benefits

This scale aims to assess how individuals perceive both the personal and societal benefits of getting vaccinated against COVID-19. It consists of three statements, each representing a different aspect of these perceptions.

Cronbach's alphas indicate satisfactory reliability of the measurement scale in Benin ($\alpha$ = 0.92) and Senegal ($\alpha$ = 0.93) [32].

## Perception of COVID-19 vaccine effectiveness

This scale aims to assess how individuals perceive the effectiveness of the COVID-19 vaccine. It consists of two statements, each representing a different aspect of this perception:

- **Personal Protection:** This variable measures individuals' belief in the likelihood of being infected with COVID-19 if they get vaccinated.

- **Risk Reduction:** This variable evaluates individuals' belief in the vaccine's ability to reduce the risk of contracting COVID-19. Responses reflect their perception of the vaccine's effectiveness in lowering the risk of getting the disease.

Cronbach's alphas indicate satisfactory reliability of the measurement scale in Benin ($\alpha$ = 0.72) and Senegal ($\alpha$ = 0.88) [32].

## Social influence on COVID-19 vaccination decision

This scale assesses the perceived social influence on the decision to receive the COVID-19 vaccine. It comprises three statements, each representing a distinct facet of this social influence:

- **Opinion of significant others**: This variable gauges individuals' perception that when the COVID-19 vaccine becomes available, most important individuals in their life (e.g., family and friends) would believe that they should receive the vaccine. Responses reflect the influence of these individuals' opinions on the decision to get vaccinated.

- **Approval of influential individuals**: This variable evaluates individuals' perception that individuals whose opinions hold significance to them would endorse the choice to get vaccinated against COVID-19 when the vaccine is offered. It quantifies the impact of these individuals' approval on the vaccination decision.

- **Healthcare personnel's perspective:** This variable measures individuals' perception that healthcare professionals would consider it advisable for them to get vaccinated for COVID-19 when the vaccine is accessible. It assesses the influence of medical staff opinions on the decision to get vaccinated.

Cronbach's alphas indicate satisfactory reliability of the measurement scale in Benin ($\alpha$ = 0.85) and Senegal ($\alpha$ = 0.87) [32].

## Behavioral control

This scale assesses individuals' perceptions of their ease of access and personal autonomy in getting vaccinated against COVID-19. It comprises four statements, each representing a different aspect of access and decision-making:

- **Access to healthcare professional:** This variable measures individuals' belief that it will be easy for them to access a healthcare professional for COVID-19 vaccination if they choose to. It reflects their perception of ease of access to vaccination services.

- **Trust in healthcare providers:** This inquiry aims to assess an individual's willingness to place their trust in healthcare professionals who would be responsible for administering a COVID-19 vaccine.

- **Access during vaccination campaigns**: This variable evaluates individuals' perception that it will be easy for them to get vaccinated against COVID-19 during organized vaccination campaigns if they wish to do so. It assesses their perception of access during mass vaccination efforts.

- **Personal Freedom to Vaccinate**: This variable gauges individuals' belief that they will have complete freedom to get vaccinated if they decide to. It reflects their perception of personal autonomy in the vaccination decision.

- **Personal Decision-Making**: This variable measures individuals' belief that it's up to them to decide whether they want to receive a COVID-19 vaccine. It emphasizes their perception of having control over the vaccination decision.

Initially, the Cronbach's alphas indicated satisfactory reliability of the measurement for Benin ($\alpha = 0.74$) and insufficient for Senegal ($\alpha = 0.52$) ([Table 2]). However, the table below shows that, for Senegal, the variable 'Personal Decision Making' is the least contributing to the total score (Item-Test Correlation) and in disagreement with the rest of the scale (Item-Rest Correlation). After excluding this item, reliability became satisfactory for Benin ($\alpha = 0.79$) and acceptable for Senegal ($\alpha = 0.60$).

## Attitudes toward COVID-19 vaccination

This scale aims to assess individuals' attitudes toward COVID-19 vaccination. It consists of five statements, each representing a different aspect of these attitudes:

- **Importance of vaccination:** This variable measures individuals' perception that it is important to get vaccinated against COVID-19. It evaluates the significance placed on vaccination as a protective measure.

- **Utility of vaccination**: This variable evaluates individuals' belief in the utility of vaccination to protect against COVID-19. It reflects the perception of the effectiveness of vaccination.

- **Responsibility for vaccination**: This variable assesses individuals' perception that it is responsible to get vaccinated against COVID-19. It evaluates the sense of duty to the community.

- **Safety of the future vaccine:** This variable evaluates individuals' belief that the future COVID-19 vaccine will not pose a health risk. It reflects the perception of the vaccine's safety.

**Table 2. Assessment of the impact of items on internal consistency.**

| Item label | Item test correlation | | Item-rest correlation | |
|---|---|---|---|---|
| | **Benin** | **Senegal** | **Benin** | **Senegal** |
| Access to healthcare professional | 0.79 | 0.59 | 0.64 | 0.31 |
| Trust in healthcare providers | 0.78 | 0.68 | 0.60 | 0.38 |
| Access during vaccination campaigns: | 0.78 | 0.73 | 0.62 | 0.48 |
| Personal Freedom to Vaccinate: | 0.72 | 0.60 | 0.57 | 0.37 |
| Personal Decision-Making | 0.59 | 0.44 | 0.43 | 0.20 |

▪ **Desirability of vaccination**: This variable measures individuals' perception of the desirability of getting vaccinated against COVID-19. It evaluates the degree of desirability associated with vaccination.

Cronbach's alphas indicate satisfactory reliability of the measurement scale in Benin ($\alpha$ = 0.82) and Senegal ($\alpha$ = 0.75) [33] (Table 3).

## Data collection and management

A system utilizing the Random Digit Dialing (RDD) method [33] was implemented to collect the data. The first step involved generating random numbers based on the market shares of various operators in each country. Valid numbers were identified by sending mass SMS messages and analyzing SMS delivery status. An automated call center then dialed the valid numbers, obtained participant consent, and completed the digital questionnaire using ODK software.

After 3 days of training on the protocol and survey content, interviews were conducted by enumerators fluent in the major national languages of Benin (Fon, Yoruba, Bariba, Dendi, and Adja) and Senegal (Wolof, Pulaar, Serer, Mandingo, and Diola) in addition to French.

Data quality assurance (DQA) spanned all phases: before, during, and after data collection. Pre-collection DQA involved tool verification, pre-testing, enumerator selection and training, and ethics approvals to enable efficient aligned data collection. During collection, DQA focused on resolving unforeseen issues, providing guidance to adapt approaches. Post-collection DQA encompassed data alignment, sorting to detect anomalies, statistical summaries, before-after comparisons, and outlier detection using graphical and statistical methods.

Prior to data sharing, all identifying information like geographic location, names, phone numbers is removed to maximize respondent confidentiality. Only domain identification codes are retained in the electronic data files.

## Data analysis

The structural equation modeling process closely adhered to the methodology outlined by Schumacker and Lomax [34] and performed in two steps using "lavaan" package [35].

Firstly, an exploratory factor analysis (EFA) was conducted to identify and explore the initial relationships between the measurement scales [36, 37]. In other words, the EFA allowed for the formulation of hypotheses about how the scales would be associated with each other in the structural equation model [36, 37].

The appropriateness of the EFA was assessed using the Kaiser-Meyer-Olkin (KMO) index (KMO>0.7), and the presence of a significant correlation structure, which is desirable for an EFA, was assessed using Bartlett's sphericity test [38] (Table 4).

The factor loading analysis suggests different relationships between the measurement scales depending on the country. Indeed, scales with factor loadings exceeding the threshold of 0.7

**Table 3. Scales, subscales, and internal consistency.**

| Scales | Subscales | Cronbach's alpha (Senegal) | Cronbach's alpha (Benin) |
|---|---|---|---|
| **Attitudes Toward COVID-19 Vaccination** | I think it is important to get vaccinated | 0.75 | 0.82 |
| | I think it is useful to be vaccinated to protect against COVID-19 | | |
| | I think it is responsible to be vaccinated against COVID-19 | | |
| | I believe that the future COVID-19 vaccine will not pose a health risk | | |
| | I think it is advisable to be vaccinated against COVID-19 | | |
| **Perceptions of COVID-19 vaccine benefits** | Getting the COVID-19 vaccine will help protect me from the virus | 0.93 | 0.92 |
| | Getting vaccinated will help fight the spread of coronavirus | | |
| | Getting the COVID-19 vaccine will help protect my loved ones from the virus | | |
| **Perception of COVID-19 vaccine safety** | I believe that those who will create the COVID-19 vaccine will ensure its safety (reverse score) | 0.65 | 0.70 |
| | Coronavirus vaccine could put my health at risk | | |
| | Coronavirus vaccine may have side effects | | |
| **Behavioral control** | I think it will be easy for me to access the health care provider to get the coronavirus vaccine if I want it | 0.60 | 0.79 |
| | How much would you trust the health care providers who would give you a COVID-19 vaccine? Would you say you trust them… | | |
| | It will be easy for me to be vaccinated against the coronavirus if I wish it during the vaccination campaigns that will be organized | | |
| | I will be completely free to get vaccinated | | |
| | It's up to me to decide if I want to get a coronavirus vaccine | | |
| **Vaccine Information Seeking** | Over the next few months, I will be learning more about the COVID-19 vaccine | 0.68 | 0.72 |
| | I will look for information on the coronavirus vaccine to better understand it. | | |
| | I will read the information I receive about the COVID-19 vaccine through social networks | | |
| **Perception of COVID-19 vaccine effectiveness** | I think that if I get the COVID-19 vaccine, it is unlikely that I will be infected | 0.88 | 0.72 |
| | I think the vaccine will reduce the risk of having COVID-19 | | |
| **Social Influence on COVID-19 vaccination decision** | When the vaccine is offered, most of the people important to me (family, friends) would think that I should get it | 0.87 | 0.85 |
| | When the vaccine is offered, the people whose opinions are important to me would approve of getting the coronavirus vaccine | | |
| | When the vaccine is offered, the nursing staff would think that I need to be vaccinated against COVID | | |

are considered, when the reliability or validity of measures is crucial, to be associated [39]. Taking these relationships into account leads us to propose a different structural model for each country (Figs 1 and 2).

**Table 4. Factors analysis results.**

| Scales | KMO's Index | | Factor loadings | |
|---|---|---|---|---|
| | Benin | Senegal | Benin | Senegal |
| Attitudes Toward COVID-19 Vaccine | 0.88 | 0.90 | 0.75 | 0.75 |
| Behavioral control | 0.87 | 0.92 | 0.75 | 0.61 |
| Perceptions of COVID-19 vaccine benefits | 0.84 | 0.85 | 0.83 | 0.88 |
| Perception of COVID-19 vaccine effectiveness | 0.87 | 0.89 | 0.67 | 0.82 |
| Perception of COVID-19 vaccine safety | 0.89 | 0.93 | -0.50 | -0.66 |
| Social Influence on COVID-19 vaccine decision | 0.89 | 0.92 | 0.71 | 0.77 |
| Vaccine Information Seeking | 0.87 | 0.76 | 0.41 | 0.25 |

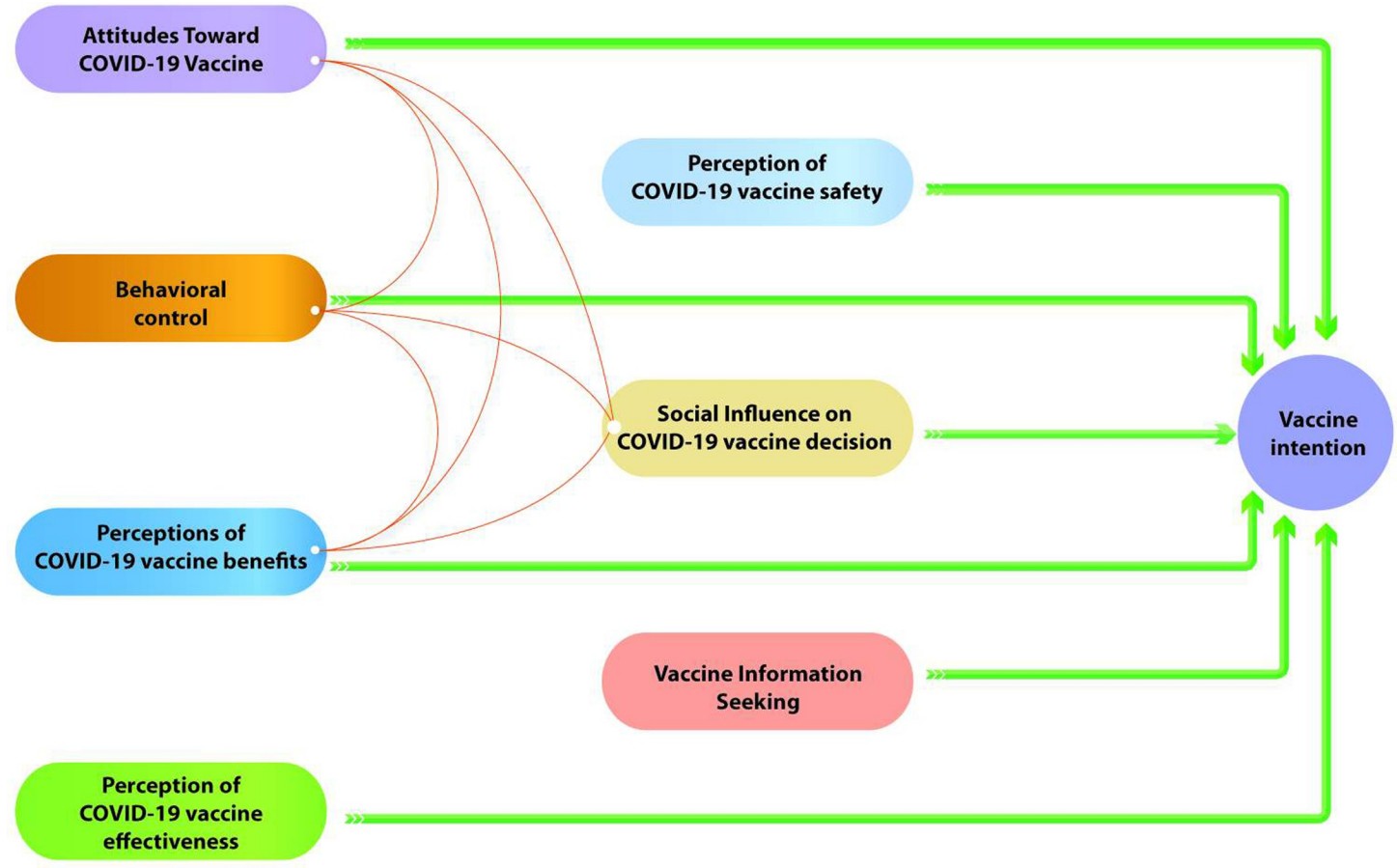

**Fig 1. Structural equation model proposed for Benin.**

Secondarily, a SEM approach was used to test and validate this theoretical model that specifies the relationships among these latent factors [40, 41]. In essence, the SEM model allowed for examining causal relationships between latent factors and observed variables, identifying potential paths of causality, and assessing the model's goodness of fit to the data [40, 41].

Regarding the estimation of SEM model parameters, several research studies have shown that applying the maximum likelihood method to ordered categorical data (Likert scales) can lead to biased estimates, inaccurate standard errors, and a misleading chi-square statistic [42, 43]. To account for the nature of the data, the diagonally weighted least squares (DWLS) estimator, the most common method, was adopted in this research [44].

For model validation, several fit indices were examined (Table 5). Overall, these indices suggest that both SEM models in Benin and Senegal demonstrate a good fit to the observed data. Indeed, the high values of CFI, TLI, NNFI, GFI, AGFI, MFI, and the low values of RMSEA and SRMR indicate a global adequacy of the model [45, 46].

## Ethical considerations

The research received approval from the National Health Research Ethics Committee of Senegal (SEN/20/23) and the Local Ethics Committee for Biomedical Research of the University of Parakou in Benin (0308/CLERB-UP/P/SP/R/SA). All individuals were informed of the ethical issues and the possibility of withdrawing from the study at any time. They all consented to participate.

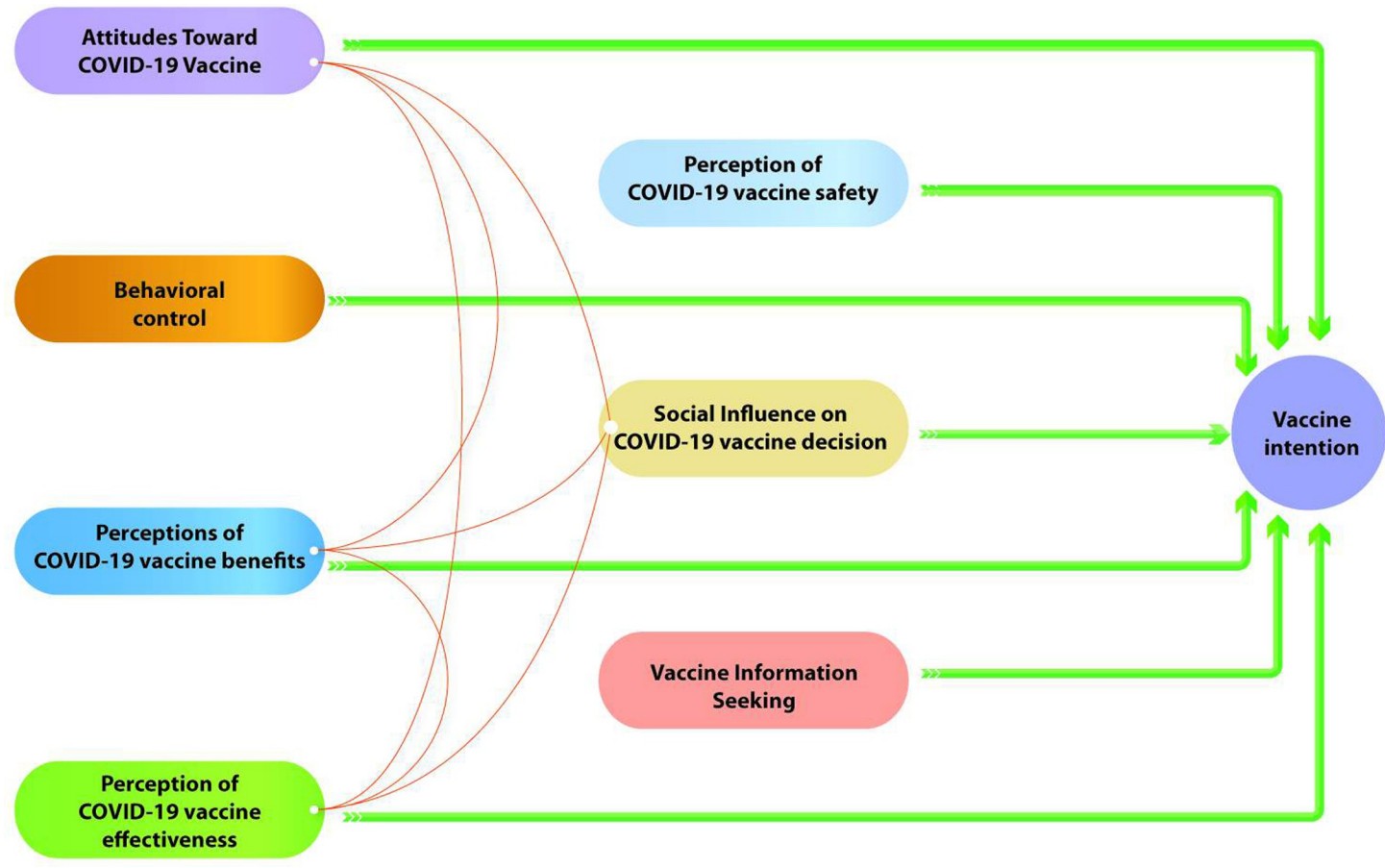

**Fig 2. Structural equation model proposed for Senegal.**

## Inclusivity in global research

Additional information regarding the ethical, cultural, and scientific considerations specific to inclusivity in global research is included in S1 Checklist.

**Table 5. Fit indicators of models.**

| Fit Indicator | Benin Model | Senegal Model |
|---|---|---|
| Comparative Fit Index (CFI) | 0.998 | 0.999 |
| Tucker-Lewis Index (TLI) | 0.986 | 0.998 |
| Non-normed Fit Index (NNFI) | 0.986 | 0.998 |
| Root Mean Square Error (RMSEA) | 0.034 | 0.013 |
| RMSEA CI (Confident intervalle) | [0.029, 0.039] | [0.000, 0.022] |
| Standardized Root Mean Square Residual (SRMR) | 0.052 | 0.042 |
| Goodness of Fit Index (GFI) | 0.983 | 0.991 |
| Adjusted Goodness of Fit Index (AGFI) | 0.977 | 0.988 |
| Parsimony Goodness of Fit Index (PGFI) | 0.737 | 0.743 |
| McDonald's Fit Index (MFI) | 0.879 | 0.980 |

## Results

### Characteristics of study sample

The age ranges from 25 to 59 years, was the majority in Senegal (67.1%), while the age range under 25 years was the majority in Benin (57.6%). Men predominated with 60.3% in Senegal and 59.2% in Benin.

In Senegal, 54.1% of men expressed a vaccination intention, while 54.8% of women did, whereas in Benin, it is higher, with 68.4% of men and 59.5% of women expressing their intention to be vaccinated (Table 6). In both countries, women appear to have a slightly lower vaccination intention compared to men, although this varies from one country to another. Indeed, overlapping confidence intervals suggest that there is a significant difference in intention to vaccinate between men and women, with a higher intention observed among men.

According to age category, the results indicate that in Senegal, vaccination intention is lowest among those under 25 (50.0%) but increases with age, reaching 63.3% among individuals aged 60 and older (Table 6). Similarly, in Benin, it is also lowest among those under 25 (65.1%) and increases with age, reaching 60.4% among individuals aged 60 and older. In both countries, there is a general trend of higher vaccination intentions among older age groups.

According to education level, it is evident that in Senegal, vaccination intention varies based on education level, ranging from 46.8% among individuals with higher education to 57.3% among those with no formal education (Table 6). In Benin, a similar trend is observed, with vaccination intention ranging from 46.1% among individuals with higher education to 70.5% among those with no formal education. In both countries, individuals with no formal education tend to have a higher vaccination intention, while those with higher education have a slightly lower intention.

### Factors promoting vaccine intention

Our analyses, based on structural equation modeling (SEM) to assess the intention to get vaccinated against COVID-19, revealed interesting nuances between two West African countries: Benin and Senegal.

In Benin, social influence ($\beta$ = 0.42, p = 0.003) and the perception of vaccine safety ($\beta$ = -0.53, p<0.001) emerge as significant determinants of vaccination intention (Fig 3). This suggests that individuals who are socially influenced are more inclined to have the intention to get vaccinated. Conversely, those who perceive the vaccine as less safe are less likely to have the intention to get vaccinated, emphasizing the importance of trust in vaccine safety.

In Senegal, positive attitudes towards COVID-19 vaccination ($\beta$ = 0.65, p = 0.013) and social influence ($\beta$ = 0.25, p = 0.048) are key determinants of vaccination intention (Fig 4). Individuals with favorable attitudes towards vaccination are more inclined to have the intention to get vaccinated, highlighting the importance of individual beliefs and personal perception of vaccination. Additionally, social influence plays a significant role, suggesting that social interactions and positive social norms surrounding vaccination also promote vaccination intention.

It is noteworthy that in both countries, positive attitudes towards COVID-19 vaccination and, perception of vaccination benefits are all significantly associated with greater social influence on vaccination decision (Figs 3 and 4). This underscores the importance of effective communication regarding vaccine benefits, as well as the promotion of positive vaccination attitudes to encourage the adoption of COVID-19 vaccination. However, despite these similarities, social influence interacts differently with other perceptions depending on the country.

**Table 6. Description of study sample.**

| Characteristics | Senegal | | | | Benin | | | |
|---|---|---|---|---|---|---|---|---|
| | Headcount | Proportion | | [95% CI] | Headcount | Proportion | | [95% CI] |
| Male | 366 | 54.1% | 49.0% | 59.2% | 350 | 68.4% | 64.2% | 72.3% |
| Female | 241 | 54.8% | 48.4% | 61.0% | 210 | 59.5% | 54.3% | 64.5% |
| < 25 years | 140 | 50.0% | 41.8% | 58.2% | 324 | 65.1% | 60.8% | 69.1% |
| 25–59 years | 407 | 54.5% | 49.7% | 59.3% | 207 | 64.9% | 59.5% | 69.9% |
| > = 60 years | 60 | 63.3% | 50.5% | 74.5% | 29 | 60.4% | 46.1% | 73.2% |
| No education | 253 | 57.3% | 51.1% | 63.3% | 98 | 70.5% | 62.4% | 77.5% |
| Primary | 122 | 56.6% | 47.6% | 65.1% | 128 | 71.1% | 64.1% | 77.3% |
| Secondary | 153 | 51.6% | 43.7% | 59.5% | 229 | 65.8% | 60.6% | 70.6% |
| Tertiary | 79 | 46.8% | 36.1% | 57.8% | 105 | 53.0% | 46.1% | 59.9% |

Indeed, it is accentuated by behavioral control in Benin (($\beta = 0.32$, p = 0.000) and the perception of vaccine efficacy in Senegal ($\beta = 0.073$, p = 0.048).

Attitudes towards the COVID-19 vaccine and perceptions of its benefits appear to play a pivotal role in vaccination-related decisions in both countries, yet they manifest in distinct ways. In Benin, attitudes towards the COVID-19 vaccine and the perceived benefits thereof seem to have a direct and significant connection with behavioral control. This suggests that in

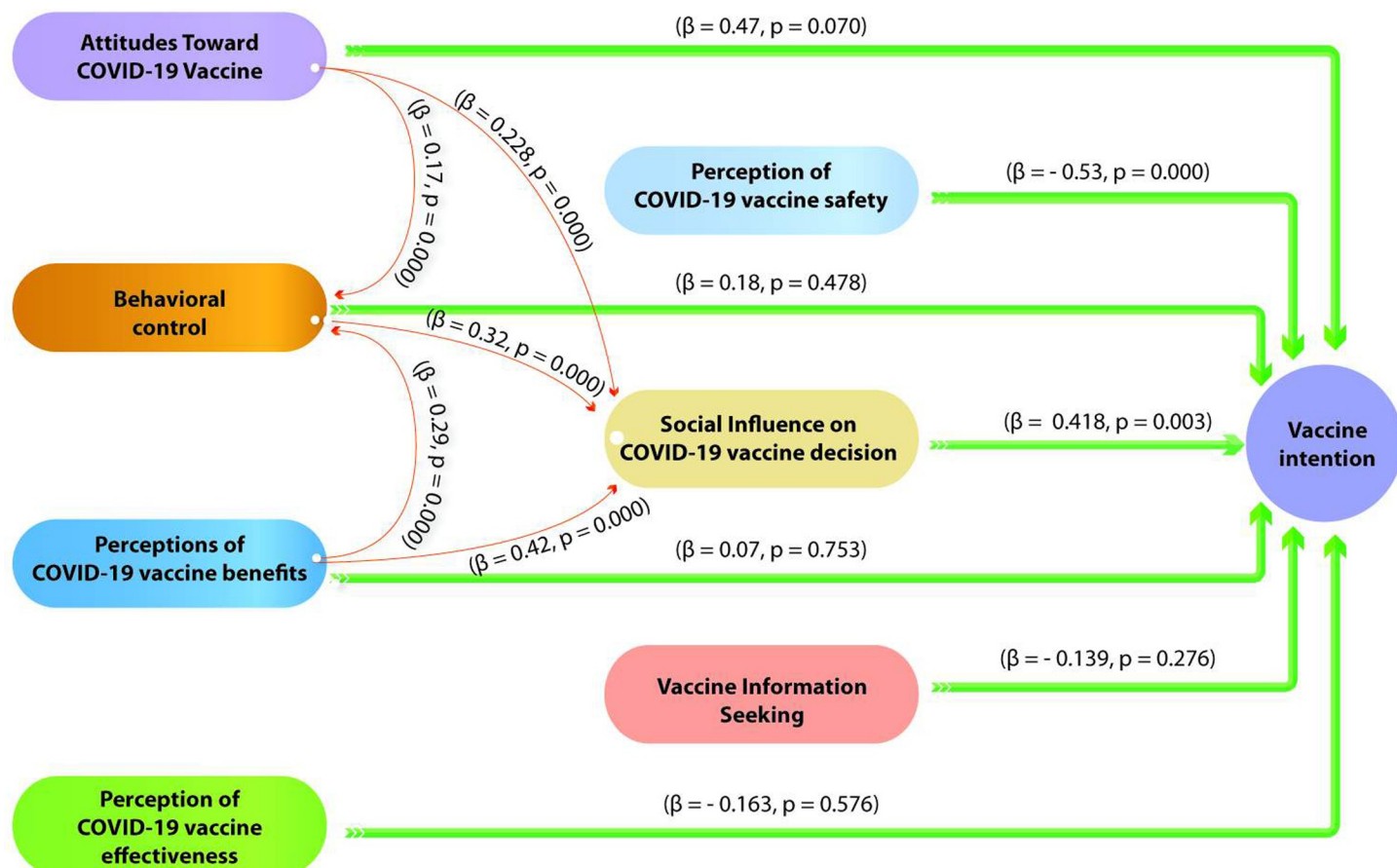

**Fig 3. Structural equation model validated for Benin.**

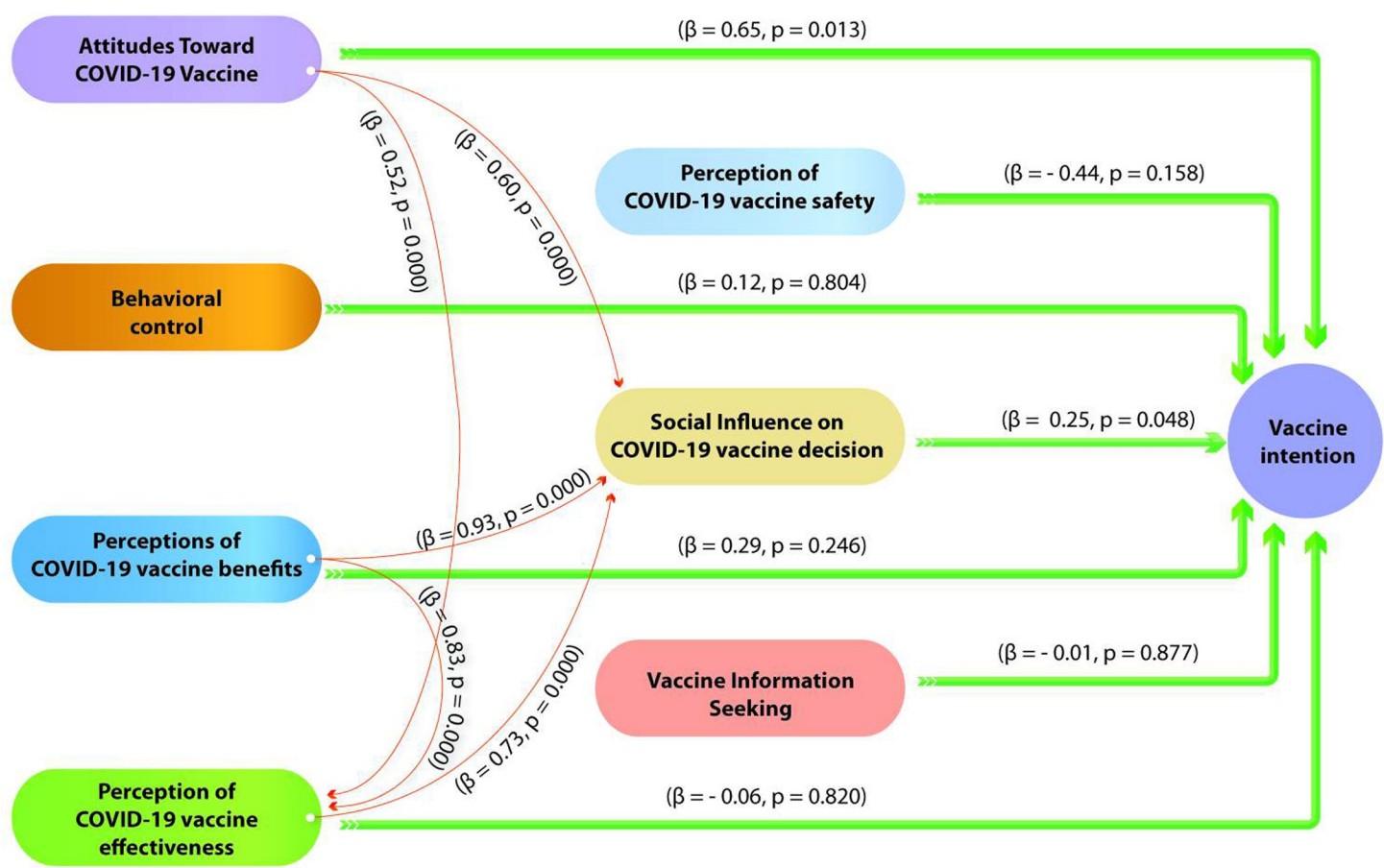

**Fig 4. Structural equation model validated for Senegal.**

Benin, the way individuals perceive the vaccine, and its advantages influences their sense of control and ability to get vaccinated. Conversely, in Senegal, these very attitudes and perceptions are closely tied to perceptions of vaccine efficacy. This implies that for Senegalese individuals, positive attitudes towards the vaccine and acknowledgment of its benefits are strongly related to their belief in its efficacy in preventing the disease. These disparities underscore the unique cultural and social contexts of the two countries, emphasizing the importance of tailored communication strategies for each setting.

In summary, although key factors vary slightly between Benin and Senegal, social influence and individual attitudes towards vaccination remain important determinants of the intention to get vaccinated against COVID-19 in both countries.

## Discussion

The analysis of determinants of COVID-19 vaccination intent using SEM for Benin and Senegal provides fascinating insight into the socio-cultural complexities of public health in West Africa.

In Benin, the **perception of vaccine safety**, a key determinant of vaccination intent, aligns with global findings highlighting the paramount importance of trust in vaccine safety [47]. Indeed, negative perceptions of vaccine safety can be exacerbated by online misinformation, an issue that has particularly intensified during the COVID-19 pandemic [48]. Furthermore,

the unprecedented speed of COVID-19 vaccine development, while globally commended, has also raised concerns [49]. These concerns underscore the need for absolute transparency about the vaccine's development process and potential side effects. Such transparency is crucial for awareness campaigns, especially at a time when online misinformation, magnified during the COVID-19 pandemic, can adversely influence perceptions of vaccine safety [48]. Moreover, to shape these perceptions, recommendations from reputable organizations are vital. For instance, the Advisory Committee on Immunization Practices' recommendation regarding the Pfizer-BioNTech COVID-19 vaccine has had a significant impact [49]. The positive influence of such recommendations underscores the importance of clear and transparent communication with the public.

**Positive individual attitudes towards vaccination**, as observed in Senegal, have previously been identified as foundational for vaccination decision-making. Brewer, N.T., et al. (2017) [50] demonstrated that applying psychological principles to the domain of vaccination could lead to more effective interventions. Furthermore, the significance of positive personal attitudes towards COVID-19 vaccination reaffirms earlier findings regarding the role of individual beliefs in vaccination intent [51]. Galanis PA, et al. (2020) [52] suggest that when attitudes towards vaccination are positive, they can be leveraged to bolster vaccination campaigns. Positive receptivity can act as a multiplier effect, where a convinced individual might influence their peers, thereby creating a network effect. Consequently, understanding and targeting these individual attitudes is crucial for crafting effective engagement strategies, as heightened receptivity to vaccines, when positive, can serve as a lever to enhance vaccination coverage [52]. Additionally, as previously mentioned by Omer, S.B., et al. (2019) [53], engaging local opinion leaders, be they religious, cultural, or media figures, can amplify these positive attitudes due to their influence and reach within their respective communities.

The primacy of **social influence** on vaccination intent observed in Benin and Senegal underscores the deep-seated role of social norms in health decision-making [54]. In many cultures, collective opinion can exert a stronger influence on individual behaviors than objective information itself [55]. This phenomenon can be attributed to individuals often prioritizing social approval and seeking to avoid stigmatization or ostracism. Moreover, it is well-established that individuals' attitudes towards vaccination are not solely shaped by personal understanding but are also heavily influenced by the opinions and attitudes of their social circles. These social influences can stem from various sources, whether it be family, friends, colleagues, or even media [50]. Additionally, religious leaders, given their stature and outreach, can mold perceptions and beliefs surrounding vaccination, either endorsing or discrediting it [53]. In certain communities, the endorsements, or resistances of religious leaders towards vaccination might carry more weight than advice from health professionals [56]. This highlights the critical importance of engaging key opinion leaders, not just for disseminating accurate and current information but also for building and sustaining public trust. An effective strategy would involve collaborating with these leaders to craft messages tailored to local culture and beliefs, ensuring broader vaccine acceptance.

A cross-country observation for both nations is the pronounced association between positive attitudes towards the vaccine and, the perception of its benefits with increased social influence. This reinforces the notion that messages about vaccine benefits are not only vital for informing but also for shaping the social landscape and norms surrounding vaccination. Campaigns that incorporate testimonials from vaccinated individuals, especially if they are esteemed or influential within their communities, might prove particularly effective [56]. This role of social norms in shaping vaccination intent has also been underscored by the identification of the "5C": confidence, complacency, constraints, calculation, and collective consciousness as psychological antecedents of vaccination [57].

In conclusion, promoting vaccination intent in West African countries will require special attention to social influence and perceptions of vaccine safety and positives attitudes. Effective, evidence-based communication will be crucial in addressing these concerns and bolstering trust in vaccination campaigns. A close collaboration with community and religious leaders will also be essential in enhancing vaccine coverage [53].

## Limits

The samples were only national representatives and did not allow for disaggregation by residence or region. Only people with a cell phone were interviewed, thus excluding the most marginalized populations. Also, having at least three questions for each scale is preferable, as is the case for all the dimensions considered except for perception of COVID-19 vaccine effectiveness.

## Conclusion

Individual perceptions and social norms profoundly influence vaccination intent against COVID-19 in Benin and Senegal. In Benin, the perceived safety of the vaccine is paramount, whereas in Senegal, positive attitudes towards vaccination prevail. These trends underscore the critical importance of combating misinformation, promoting transparency, and closely collaborating with local opinion leaders to strengthen trust in vaccination campaigns. It would thus be relevant to delve deeper into how local media and social networks shape vaccination perceptions, to define targeted strategies for each community, and to further explore the impact of religious and community leaders on vaccine acceptance. Moreover, given the significance of social norms in health decision-making, a more in-depth study of the underlying social mechanisms could provide invaluable insights for future campaigns.

## Supporting information

**S1 Checklist. Inclusivity in global research.**
(DOCX)

## Author Contributions

**Conceptualization:** Ibrahima Gaye, Valery Ridde, Adama Faye.

**Data curation:** Ibrahima Gaye.

**Funding acquisition:** Valery Ridde, Adama Faye.

**Investigation:** Ibrahima Gaye, Valery Ridde, Adama Faye.

**Methodology:** Ibrahima Gaye, Adama Faye.

**Project administration:** Valery Ridde, Adama Faye.

**Software:** Ibrahima Gaye.

**Validation:** Ibrahima Gaye.

**Visualization:** Ibrahima Gaye.

**Writing – original draft:** Ibrahima Gaye.

**Writing – review & editing:** Valery Ridde, Elías Martinien Avahoundjea, Mouhamadou Faly Ba, Jean-Paul Dossou, Amadou Ibra Diallo, Adama Faye.

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
