## [Decision Letter · Decision Letter 0]

12 Dec 2023

PGPH-D-23-02275

Cross-sectional study on Intention to be vaccinated against COVID-19 in Benin and Senegal: a structural equation modelling (SEM)

Dear Ridde,

Thank you for submitting your manuscript to PLOS Global Public Health. After careful consideration, we feel that it has merit but does not fully meet PLOS Global Public Health’s publication criteria as it currently stands. Therefore, we invite you to submit a revised version of the manuscript that addresses the points raised during the review process.

We look forward to receiving your revised manuscript.

Kind regards,

Collins Otieno Asweto, PhD

Academic Editor

Journal Requirements:

2. Please include a complete copy of PLOS’ questionnaire on inclusivity in global research in your revised manuscript. Our policy for research in this area aims to improve transparency in the reporting of research performed outside of researchers’ own country or community. The policy applies to researchers who have travelled to a different country to conduct research, research with Indigenous populations or their lands, and research on cultural artefacts. The questionnaire can also be requested at the journal’s discretion for any other submissions, even if these conditions are not met.  Please find more information on the policy and a link to download a blank copy of the questionnaire here: https://journals.plos.org/globalpublichealth/s/best-practices-in-research-reporting. Please upload a completed version of your questionnaire as Supporting Information when you resubmit your manuscript.

3. Please amend your detailed Financial Disclosure statement. This is published with the article. It must therefore be completed in full sentences and contain the exact wording you wish to be published.

a. State what role the funders took in the study. If the funders had no role in your study, please state: “The funders had no role in study design, data collection and analysis, decision to publish, or preparation of the manuscript.”

b. If any authors received a salary from any of your funders, please state which authors and which funders.

4. We have noticed that you have uploaded Supporting Information files, but you have not included a list 3of legends. Please add a full list of legends for your Supporting Information files after the references list.

5. In the online submission form, you indicated that "Data will be available after a request". All PLOS journals now require all data underlying the findings described in their manuscript to be freely available to other researchers, either 1. In a public repository, 2. Within the manuscript itself, or 3. Uploaded as supplementary information.

Reviewer's Responses to Questions

**Comments to the Author**

1. Does this manuscript meet PLOS Global Public Health’s publication criteria? Is the manuscript technically sound, and do the data support the conclusions? The manuscript must describe methodologically and ethically rigorous research with conclusions that are appropriately drawn based on the data presented.

Reviewer #1: Yes

Reviewer #2: Yes

Reviewer #3: Yes

Reviewer #4: Yes

2. Has the statistical analysis been performed appropriately and rigorously?

Reviewer #1: Yes

Reviewer #2: Yes

Reviewer #3: Yes

Reviewer #4: Yes

3. Have the authors made all data underlying the findings in their manuscript fully available (please refer to the Data Availability Statement at the start of the manuscript PDF file)?

Reviewer #1: No

Reviewer #2: No

Reviewer #3: No

Reviewer #4: Yes

4. Is the manuscript presented in an intelligible fashion and written in standard English?

Reviewer #1: Yes

Reviewer #2: Yes

Reviewer #3: Yes

Reviewer #4: Yes

5. Review Comments to the Author

Reviewer #1: In this paper, the authors conducted a cross-sectional study by surveying 865 Beninese and 813 Senegalese through random phone dialing. The survey examined seven key factors that may influence the respondent’s intention to get vaccinated against COVID-19. The analysis utilizes a structural equation modeling approach, which is common in understanding the causal paths between different potential factors that may impact the outcome.

The study draws on both the similarity and the difference between the key factors identified that are the most related with the outcome (intention to vaccination) through the survey and analysis. For example, the authors found the vaccination intention to be driven by perception of vaccine safety in Benin but attitudes towards the vaccine in Senegal. However, more work is needed in the interpretations of the results and other key factors.

Overall, the manuscript was easy to read and follow. The statistics and results were well summarized into tables and graphs, but descriptions for methods and/or choices of certain parameters are missing in some exhibits.

Major concerns:

1. In the questionnaire, was the respondent being asked about their current vaccine status? Those who had already been vaccinated (with at least one dose) will likely have different perspectives to many of the questions compared to those who “intended” to get vaccinated.

2. Line 115 – 116 under Methods, given how the quota sampling method was used, a multilevel SEM may be able to better address the correlation between, for example, respondents from the same region – because the survey could possibly reach many people within close communities by random dialing stratified by region.

3. In the SEM analysis, a connection/effect was not observed between attitude and perception of vaccine efficacy in Benin because you didn't allow for such a causal path to exist in the proposed model structure? Similar for the causal path between attitude and behavioural control in Senegal, this path was not allowed in the proposed model structure. From my understanding of how the lavaan package works, if you did not specify the causal path, then it will assume the two items are independent by design?

4. In discussion, while we don't see the model results give us a significant connection between perception of vaccine safety and vaccination intent in Senegal, how should we interpret this result and what are its implications? For instance, does this suggest that for the communication to be more effective, Senegal should focus less on communicating the safety/risk of vaccine because it is not a dominant factor, or is it because this result may be confounded with other factors such as personal attitude in the analysis? In general, more work needs to be done in communicating/interpreting the results and how the factors are connected to each other.

Other minor concerns & suggestions:

1. In SEM, how did you handle the correlation between different questions in the survey? e.g., both attitude and behavioural control are significantly connected to social influence in Benin, but themselves are also significantly correlated.

2. In table 1, can you provide more details about why setting p (expected proportion) to 50%? What does this quantity mean in the context of this study and what are the implications by setting it to different values?

3. Line 120, you mentioned about post-stratification statistical analyses were performed to adjust the data from the final sample. What exact post-stratification methods did you consider and implement?

4. Line 87, a 50.2% of female in the population is far from being described as "predominantly" - it's in fact quite an even split.

5. Line 345 – 346, are there any reasons for why more educated people are less intentional to get vaccinated? For example, did higher educated people also share lower perception of vaccine safety or more negative attitudes which led to a lower intention to vaccination?

6. There is a gap of a couple months in the implementation between Senegal (Dec 2020) and Benin (March 2021). This should not be a huge concern but it’s worth mentioning any change or vaccine campaigns happening during this period that may have altered people’s perceptions.

Reviewer #2: The Manuscript meets the PLOS Global public health's publication criteria and follows a scholarly writing outline. It is a well written manuscript with good level language and scholarly tone. The title, the objectives and methods are clearly described and follows a scientific writing fashion. Appropriated ethical clearness have secured and the paper has both research and publication ethics. Data is not available, but the author clearly mentioned that the data will be available up on request.

Reviewer #3: This is a well-written manuscript. As I am not a native English speaker, my assessment of the quality of the English used in this manuscript may be inadequate. The methodology is described in detail and the conclusions are realistic.

The authors have not made public all the data required to reproduce the results of their study. It would be good if they could explain why.

Thank you.

Reviewer #4: Coronavirus disease-19 is a pathology which has evoked important discussions in health from local to international levels and it's not ending anytime soon. I am optimistic this publication will improve the advocacy and management of COVID-19 and future pandemics not just in the global south but also the global north. I commend the authors for putting this together, it is an excellent research and well written from authors that come from countries whose primary language is not English.

Please make the following corrections;

1. Full name of COVID-19 in the title.

2. Line 25, COVID-19 in full then abbreviation in bracket. Abbreviations can then be used for the rest of the publication.

3. Line 32, March 29 in which year?

4. Please capitalize all first alphabets for every word in the title.

5. Line 26, were, not was vaccinated.

6. Line 136, action (s).

7. Line 158, captures or captured?

8. Line 189, concerns not concern.

9. Line 201, measures not measure.

10. Line 211, gauges not gauge.

11. Line 225, behavioral control, capitalize first alphabet.

12. Line 340, Intentions not intention (2 countries)

13. Line 447, were only national representatives not nationally representative.

6. PLOS authors have the option to publish the peer review history of their article (what does this mean?). If published, this will include your full peer review and any attached files.

**Do you want your identity to be public for this peer review?** For information about this choice, including consent withdrawal, please see our Privacy Policy.

Reviewer #1: No

Reviewer #2: **Yes: **Solomon Chane Abera

Reviewer #3: **Yes: **Ibrahima Barry

Reviewer #4: **Yes: **AZEEZAT FAJEMBOLA

---

## [Decision Letter · Decision Letter 1]

6 Feb 2024

Cross-sectional study on Intention to be vaccinated against COVID-19 in Benin and Senegal: a structural equation modelling (SEM)

PGPH-D-23-02275R1

Dear Ridde,

We are pleased to inform you that your manuscript 'Cross-sectional study on Intention to be vaccinated against COVID-19 in Benin and Senegal: a structural equation modelling (SEM)' has been provisionally accepted for publication in PLOS Global Public Health.

Best regards,

Collins Otieno Asweto, PhD

Academic Editor

Reviewer Comments (if any, and for reference):

Reviewer's Responses to Questions

**Comments to the Author**

1. If the authors have adequately addressed your comments raised in a previous round of review and you feel that this manuscript is now acceptable for publication, you may indicate that here to bypass the “Comments to the Author” section, enter your conflict of interest statement in the “Confidential to Editor” section, and submit your "Accept" recommendation.

Reviewer #1: All comments have been addressed

Reviewer #3: All comments have been addressed

Reviewer #4: All comments have been addressed

2. Does this manuscript meet PLOS Global Public Health’s publication criteria? Is the manuscript technically sound, and do the data support the conclusions? The manuscript must describe methodologically and ethically rigorous research with conclusions that are appropriately drawn based on the data presented.

Reviewer #1: Yes

Reviewer #3: Yes

Reviewer #4: Yes

3. Has the statistical analysis been performed appropriately and rigorously?

Reviewer #1: Yes

Reviewer #3: Yes

Reviewer #4: Yes

4. Have the authors made all data underlying the findings in their manuscript fully available (please refer to the Data Availability Statement at the start of the manuscript PDF file)?

Reviewer #1: Yes

Reviewer #3: Yes

Reviewer #4: Yes

5. Is the manuscript presented in an intelligible fashion and written in standard English?

Reviewer #1: Yes

Reviewer #3: Yes

Reviewer #4: Yes

6. Review Comments to the Author

Reviewer #1: All major concerns are addressed in the revised version. It was a pleasure reading the updated manuscript!

Reviewer #3: Thanks to the authors for providing the data needed to reproduce the results of their study. These data can be accessed via this link: https://doi.org/10.6084/m9.figshare.24574432.v2.

Reviewer #4: Dear Authors,

You have done a great job. I am augmenting my previous submission with these;

1. Line 207- greater concern (s) or concerns.

2. Under limits of your research, the last line, is there any reason why perception should start with a capital letter?

Goodluck.

7. PLOS authors have the option to publish the peer review history of their article (what does this mean?). If published, this will include your full peer review and any attached files.

**Do you want your identity to be public for this peer review?** For information about this choice, including consent withdrawal, please see our Privacy Policy.

Reviewer #1: No

Reviewer #3: **Yes: **Ibrahima Barry

Reviewer #4: **Yes: **AZEEZAT FAJEMBOLA
